# Performance and Hydroponic Tomato Crop Quality Characteristics in a Novel Greenhouse Using Dye-Sensitized Solar Cell Technology for Covering Material

**Georgios K. Ntinas, Kalliopi Kadoglidou \*, Nektaria Tsivelika, Konstantinos Krommydas, Apostolos Kalivas, Parthenopi Ralli and Maria Irakli**

Hellenic Agricultural Organization-Demeter, Institute of Plant Breeding and Genetic Resources, Thermi, 57001 Thessaloniki, Greece; georgiosntinas@gmail.com (G.K.N.); riatsivel@gmail.com (N.T.); kos_krom@hotmail.com (K.K.); kalivasapostolis@yahoo.gr (A.K.); pralli@ipgrb.gr (P.R.); irakli@cerealinstitute.gr (M.I.)

\* Correspondence: kkadogli@agro.auth.gr; Tel.: +30-23-1047-1110

**Abstract:** In this study, we evaluated crop productivity and physiology during the hydroponic cultivation of medium-sized and cherry tomato crops, using two experimental greenhouses. Of the greenhouses, one used dye-sensitized solar cell (DSSC) technology for covering material, whilst the other, a conventional one (CONV), was covered using diffusion glass as a control. The effect of the colored lighting that resulted from the DSSC glass filtering on the physiological response of the crops was examined by measuring the plant transpiration rate and leaf chlorophyll content. Furthermore, we evaluated potential differences in the concentration of phytochemical compounds, such as ascorbic acid, lycopene, and quality characteristics. Tomato plants in the DSSC greenhouse presented lower early and total yields, as well as lower chlorophyll content, stomatal conductance, photosynthetic rate, and transpiration rate values, especially in the medium-sized fruits, as compared to the CONV greenhouse. The DSSC greenhouse showed significantly higher values of bioactive compounds for both the cherry and medium-sized tomato, with increases in the ascorbic acid, lycopene, β-carotene, and total carotenoids concentration, which ranged from 6% to 26%. Finally, for both the hybrids, the 2,2′-azino-bis-3-ethylbenzthiazoline-6-sulphonic acid (ABTS) and 1,1-diphenyl-2-picrylhydrazyl (DPPH) tests showed circa 10% and 5% increase, respectively, in the DSSC greenhouse.

**Keywords:** semi-transparent photovoltaic modules; hydroponics; tomato; bioactive ingredients; lycopene; antioxidant capacity

## 1. Introduction

Across the world, controlled environment agricultural systems function as a dynamic production process. Such systems provide shelter for the crops against the direct influence of external weather conditions, and they offer the opportunity to modify the indoor climate to create an environment that is optimal for crop growth and production, both in terms of quality and quantity [1,2]. Following the first energy crisis in the seventies, during which limited energy supplies led to an increase in energy prices, greenhouse energy consumption has again become a major issue. There is a significant need for energy cost reduction, since energy constitutes a substantial fraction of the total production costs.

In Mediterranean countries, energy consumption for control of environmental conditions constitutes approximately 20–30% of the total production costs, with a higher percentage in the northern countries. Moreover, due to the rising interest in climate change and greenhouse gas

emissions (GHG), the use of fossil fuels is part of the political agenda. Therefore, the greenhouse industry is facing economic, political, and social pressure to use renewable energy sources, reduce energy usage and $CO_2$ emissions, and improve greenhouse energy efficiency via technological innovations [3–5]. Different types of photovoltaic (PV) modules for greenhouse roofs to harvest solar radiation are available [6–8]. In this case, there is competition for light between PV modules and plants, which can lead to decreased production [9]. However, excess light, especially during the summer, is unnecessary for plant growth, where channeling of the excessive light toward electricity generation is an option. Furthermore, high air temperatures inside the greenhouse during summer can decrease the photosynthetic rate, and, therefore, plant growth and crop yield [10,11].

Roslan et al. [12] presented new developments in dye-sensitized solar cells (DSSC) for greenhouse shading and electricity production. The authors' reported that DSSC had not been applied in a greenhouse setting for plant growth and energy saving. The application of photoselective DSSC in greenhouse covering can regulate environmental conditions and enhance the quantitative and qualitative characteristics of greenhouse products [13].

Two important parameters of tomato (*Solanum lycopersicum* L.) quality are the pH and titratable acidity. Tomatoes are high-acid foods, such that active thermal treatments are not required for the destruction of microorganisms that contribute to food spoilage. The pH of tomato fruits ranges from 4.0 to 4.5, such that the lower the pH, the greater the so-called "tartness", a factor by which some consumers judge the quality of the tomato fruit [14]. Citric acid is the basic acid found in tomatoes and it contributes to the titratable acidity. Total soluble solids (°Brix) and °Brix / titratable acidity are the common indicators that express the taste of a tomato. °Brix values are measured using a refractometer and they indicate the percentage (%) of the dissolved solids in a solution, which in tomato paste is mainly the total sugars (glucose, fructose), acids (citric and malic), and other components in a lower proportion (phenols, amino acids, ascorbic acid, and inorganic salts) [15].

Tomatoes are a good source of nutrients and bioactive compounds, such as carotenoids (lycopene, β-carotene, and lutein), vitamin C, and polyphenolic compounds, which are thought to be health-promoting factors with antioxidant properties [16,17]. The nature and concentration of these compounds depends on the cultivation practices, environmental factors, crop variety, and maturity [18]. There is a great deal of variation between crop varieties in terms of fruit size and color, which can affect the nutritional properties of tomato. Lenucci et al. [19] found significant differences among 14 varieties of cherry tomato (ChT) and four medium-sized tomato (MST) hybrids in terms of the lycopene and β-carotene content, where varieties with more pigments had a higher concentration of lycopene. Consumption of tomato and tomato products has been associated with a lower risk of developing digestive problems and prostate cancer.

The lycopene content depends on the redness of the tomato fruit, and it is the main carotenoid and one of the key antioxidants found in fresh tomatoes and processed tomato products. Vitamin C, including ascorbic and dehydroascorbic acid, is important for the protection of tomatoes from the auto-oxidizing factors that may increase when ripening. The role of ascorbic acid in the prevention of diseases associated with oxidative damage occurs because of its ability to neutralize the action of free radicals in biological systems [20]. In addition to its antioxidant action, ascorbic acid is essential to life because of its many physiological effects. Plants and a vast majority of mammals, but not humans, are able to synthesize it. Meanwhile, the main sources of vitamin C are citrus fruits, tomatoes, and potatoes [21]. Ascorbic acid is relatively stable in tomatoes, due to the acidic conditions prevailing in the tissue. However, it easily decomposes due to oxidation, exposure to light, or high temperatures. Significant losses of ascorbic acid occur during the postharvest storage period. Therefore, lowering the temperature from ambient (20 °C) to cool (4 °C) or freezing (−18 °C) reduces the rate of loss of ascorbic acid.

Tomato fruit maturation is a complex process involving various morphological, physiological, biochemical, and molecular processes. These processes include the reduction of chlorophyll, the synthesis and storage of carotenoids (mainly lycopene and β-carotene) and aromatic compounds,

changes in the metabolism of organic acids, and the softening of the fruit tissue that occurs in combination with increased $CO_2$ and ethylene production by the fruit. The quantity of other important antioxidants, such as ascorbic acid and phenolic compounds, changes during the maturation of tomato [22,23]. Specifically, Ilahy et al. [22] described that the increase in antioxidants did not follow the fruit ripening pattern, while Helyes and Lugasi [23] found that lycopene and total antioxidant capacity increased during maturation, while polyphenol content remained almost the same. All of these processes affect texture, color, flavor, aroma, but also the content of antioxidant compounds and the antioxidant effect of tomato fruit [24].

Motivated by the above concept, the purpose of this study was to evaluate and quantify the gain of crop production in terms of higher crop yield and improved quality of hydroponic tomato cultivation in a greenhouse, using the innovative technology of DSSC for the glass cover. All energy harvested from the sun with the DSSC was used to cover part of the electricity consumption of the DSSC greenhouse with a renewable energy source, in comparison to a reference greenhouse that had a glass cover and conventional electricity was provided by the grid (CONV greenhouse). Specific parameters related to plant physiology, such as transpiration, stomatal conductance, photosynthetic rate, and chlorophyll content were determined. Additionally, chemical analyses of harvested fruits were conducted to determine potential differences in the concentration of phytochemical compounds, such as ascorbic acid and lycopene, and quality characteristics between the raw products harvested from the DSSC greenhouse and the CONV one.

## 2. Materials and Methods

### 2.1. Hydroponic Tomato Cultivation Set Up in the CONV and DSSC Greenhouses

The experiment was conducted in a CONV and in a DSSC greenhouse, at the Institute of Plant Breeding and Genetic Resources in Thermi Thessaloniki (22°59.956′ E/40°32.281′ N, −1 m.a.s.l.) in the summer of 2015. After preliminary assessments—resistance studies of various hybrids—two commercial table tomato hybrids, "Oasis" (MST) and "Genio" (ChT), were selected for this experiment (Figure 1).

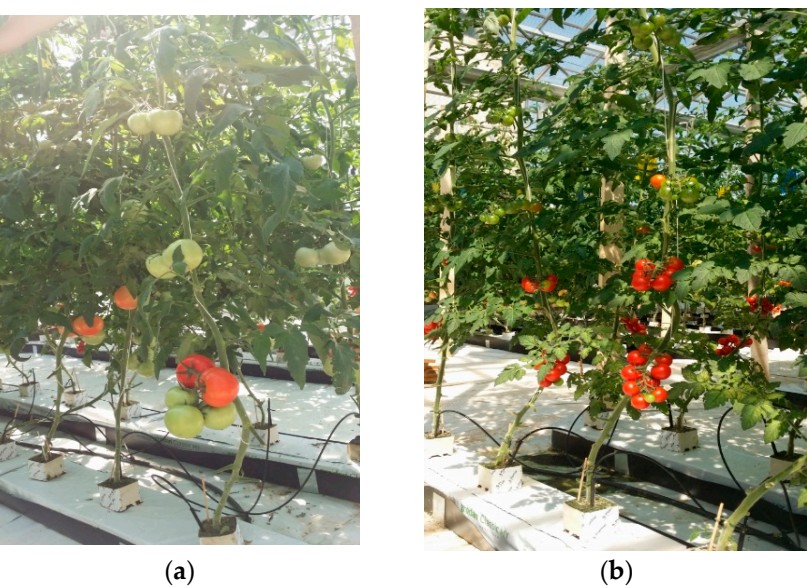

(**a**)          (**b**)

**Figure 1.** Table tomato hybrids selected for cultivation: (**a**) Hybrid "Oasis" (medium-sized tomato, MST); (**b**) hybrid "Genio" (cherry tomato, ChT).

The tomato plants were grown in double lines of 4 m length and distances of 33 cm (plant to plant) and 100 cm (double-row corridor). In total, there were nine double lines in each greenhouse,

four of which were cultivated with medium-sized tomato (MST) and the other five with cherry tomato (ChT) (Figure 2). The tomato seedlings, planted in rockwool cubes, were transplanted in their final position on rockwool substrates (Grotop Master Grodan, Denmark) when they reached the stage of 5–6 true leaves. In both greenhouses, tomato plants were fertirigated with Hoagland's solution (Table 1). A total of 2.8 L of $HNO_3$ was added in a separate tank containing 100 L of water. The pH and the EC of the nutrient solution were 5.5–6.0 and 3.0–3.5 dS m$^{-1}$, respectively. The EC and pH were measured using portable EC and pH meters (HI 8733 and HI 8424, Hanna Instruments, Inc., Woonsocket, RI, USA).

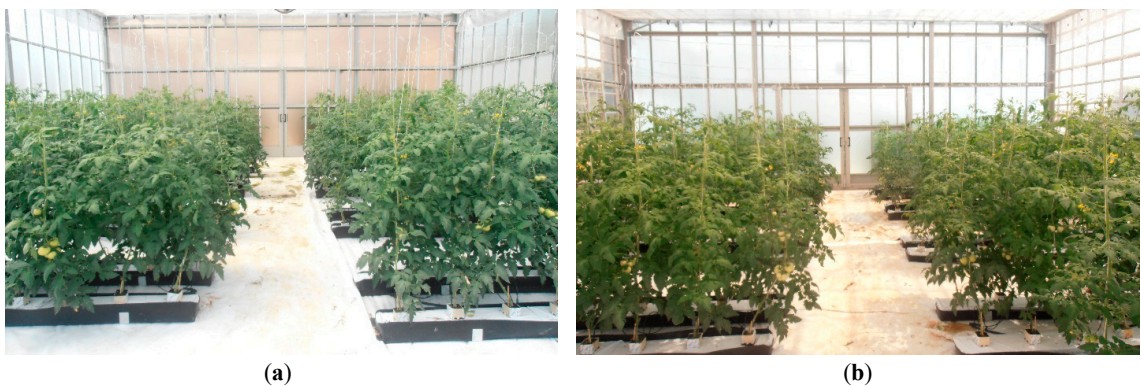

(**a**)　　　　　　　　　　　　　　　　　　　　　(**b**)

**Figure 2.** Plant arrangement in the CONV (Conventional) (**a**) and DSSC (Dye-Sensitized Solar Cell) (**b**) greenhouses. The difference in the shading within the greenhouses is visible.

**Table 1.** Chemical characteristics of the nutrient solutions used in the experiments.

| Stock Solution (Tank A) | | Stock Solution (Tank B) | | Micronutrients (Tank C) | |
|---|---|---|---|---|---|
| **Macronutrients** | **(g/100 L)** | **Macronutrients** | **(g/100 L)** | **Micronutrients** | **(g/20 L)** |
| $Ca(NO_3)_2$ | 6800 | $K_2SO_4$ | 2780 | $MnSO_4$ | 585 |
| $KNO_3$ | 1000 | $KNO_3$ | 1320 | $Na_2B_4O_7$ | 530 |
| EDTA-Fe (13%) | 200 | $MgSO_4$ | 1300 | $CuSO_4$ | 40 |
| | | $KH_2PO_4$ | 2340 | $ZnSO_4$ | 270 |
| | | | | $(NH_4)_6Mo_7O_{24}$ | 25 |

Preventive application of abamectin against *Tetranychus urticae* control and *Bacillus thuringiensis* subsp. kurstaki strain SA-11 for *Heliothis armigera* control were carried out once in the middle of the experiment.

Microclimatic parameters inside both greenhouses (solar radiation, air temperature, and humidity) were recorded with a HOBO data logger (HOBO micro-station, Onset Inc., USA).

### 2.2. Evaluation of Cultivation in the Two Greenhouses

Evaluation of the cultivation of tomatoes in the two greenhouse compartments was performed by measuring yield and morphological, physiological, and qualitative characteristics, related to chemical, nutritional, and bioactive ingredients as well as antioxidant activity.

### 2.2.1. Fruit Yield and Morphological Characteristics

Fruit were harvested weekly by collecting only ripe fruits from each greenhouse and each hybrid. The sum of the first and the second harvest represented the early yield. In total, seven harvests took place, and fruits were weighed and ranked in quality classes based on size, weight, and shape directly after harvest. The International Union for the Protection of New Varieties of Plants (UPOV) descriptor was used to measure and evaluate morphological characteristics.

### 2.2.2. Plant Physiological Parameters

Transpiration, stomatal conductance, and photosynthetic rate of plants was measured 46 days after transplanting (DAT), by using an infrared gas analyzer (LCi-SD portable photosynthesis system, ADC BioScientific Ltd., Hertfordshire, UK). For this purpose, in each twin cultivation row (24 plants), five measurements were taken from the upper third of a fully developed leaf, between the fourth and fifth inflorescence. The photosynthetically active radiation (PAR) was $576 \pm 100$ and $387 \pm 100$ µmol (photon) $m^{-2}$ $s^{-1}$ in the CONV and DSSC greenhouses, respectively. The $CO_2$ concentration in the chamber was $390 \pm 10$ µmol ($CO_2$) $mol^{-1}$, the temperature in the chamber was set at $30 \pm 2$ °C, and the water reference as partial pressure was $35 \pm 3$ mbar. Additionally, the chlorophyll content of the leaves was measured on the fortieth day using a portable Chlorophyll Content Meter (CCM-200, Opti-Sciences, Inc., Hudson, NH, USA). For this purpose, ten measurements were taken from the terminal leaflet of a fully developed leaf, between the fourth and fifth inflorescences.

### 2.2.3. Fruit Physicochemical Parameters

From each harvest, representative fruit samples from each hybrid and each greenhouse were collected at each inflorescence stage, to determine the quality parameters described below. Prior to measurements, samples were cut into pieces and homogenized in a conventional blender in order to obtain the tomato pulp. For each measurement, three representative samples of tomato fruit, from each hybrid and truss, were taken in both greenhouses and each measurement was repeated three times (three replicates). Tomato pulp pH was measured with a portable pH meter (MW802, Milwaukee Instruments Inc., Rocky Mount, NC, USA) and dry weight was determined after oven drying (48 h, 72 °C). Free sugars and citric acid in tomato pulp were determined enzymatically using the Megazyme kit of glucose/fructose/sucrose and the citric acid, respectively, according to the corresponding protocol (Megazyme International, Wicklow, Ireland).

### 2.2.4. Fruit Antioxidant Compounds

Ascorbic acid was determined by extraction of tomato pulp with 20% metaphosphoric acid and titration with 2,6-dichloroindophenol. The ascorbic acid content was calculated according to a standard and expressed as mg ascorbic acid/100 g of tomato pulp [25]. Carotenoids were extracted with a mixture of hexane/ethyl acetate (50:50, *v/v*) from tomato pulp in a ratio of 1:10 with 0.1% butylated hydroxytoluene (BHT) in an ultrasonic bath for 5 min according to Irakli et al. [26]. The procedure was repeated until discoloration of the extract was reached. The total extracts were concentrated to dry weight on a rotary evaporator and then re-dissolved in 2 mL of dichloromethane/methanol/acetonitrile solution (30/30/40, *v/v/v*). Consequently, the extracts were filtered through 0.45 µm polytetrafluoroethylene (PTFE) membrane filters and an aliquot of 20 µL of this was introduced into a high-performance liquid chromatography (HPLC) system (Agilent Technologies, series 1200, Urdorf, Switzerland,) connected to a photodiode array detector (DAD). Separation was accomplished on an YMC $C_{30}$ column ($250 \times 4.6$ mm; 5 µm) at a temperature of 15 °C. Elution was performed following a gradient elution program starting from a mixture of acetonitrile/methanol (85/15, *v/v*) at a speed of 1.5 mL/min and resulting in a mixture of acetonitrile/methanol/dichloromethane (30/20/60, *v/v/v*). The DAD was set at 450 nm for β-carotene and 475 nm for lycopene. The components were identified based on elution times and absorption spectra of reference standards.

### 2.2.5. Antioxidant Capacity

An aliquot of 2 g of tomato pulp was extracted two times with 20 mL of a 70% aqueous methanol mixture followed by ultrasonication for 15 min and centrifugation at 3000 rpm for 10 min. The supernatants were collected, homogenized, and methanolic tomato extracts were used for the evaluation of antioxidant capacity using the 2,2′-azino-bis-3-ethylbenzthiazoline-6-sulphonic acid (ABTS) and 1,1-diphenyl-2-picrylhydrazyl (DPPH) tests.

The ABTS radical scavenging activity assay was evaluated according to the protocol of Re et al. [27]. The ABTS$^{\bullet+}$ radical was produced by the oxidation of 7.4 mM ABTS (2,2′-azino-bis-3-ethylbenzothiazoline-6-sulfonic acid diammonium salt) with 2.45 mM potassium persulfate in water. The mixture was placed in the dark at room temperature for 16 h before use, and then the ABTS$^{\bullet+}$ solution was diluted with water to an absorbance of 734 nm of 0.70 ± 0.02. After the addition of 100 μL of extract to 3.9 mL of ABTS solution, the absorbance was measured against a blank at 734 nm after 4 min. The results were obtained by interpolating the absorbances on a calibration curve obtained with Trolox. The results were expressed as mg Trolox equivalents per 100 g of dry mass (mg TE/100 g).

The DPPH$^+$ radical scavenging ability assay was performed using tomato methanolic extracts mixed with DPPH$^+$ (2,2-diphenyl-1-picrylhydrazyl) reagent and absorbance was read at 517 nm according to Yen and Chen [28]. A volume of 0.15 mL extract was reacted with 2.85 mL 0.1 mM methanolic solution of DPPH. After 5 min, the absorbance at 516 nm was recorded. DPPH+ reagent was used as blank, and reduction percentage of free radical scavenging activity was monitored. Trolox was used as the standard. The results were expressed as mg Trolox equivalents (TE) per 100 g RB (mg TE/g).

*2.3. Statistical Analysis*

The statistical analysis of all experimental data (ANOVA) was carried out with the MSTAT-C version 1.41 statistical program (Michigan State University, East Lansing, MI). The experimental design was a completely randomized block (CRBD) split plot with greenhouse type (glass or DSSC) as main plots and hybrid species as subplots with five replications for the production and yield data and with six replications for the physiological parameters. For the comparison of the averages, the least significant difference criterion (Least Significant Difference Test, LSD test) was used at $P \leq 0.05$. The figures presented in the manuscript were created using an Excel spreadsheet. In each figure, the values are the average of the evaluated parameter, in both tomato hybrids (ChT and MST) and both greenhouses (CONV, DSSC). The averages of a parameter followed by different letters in the same column were statistically significantly different ($P < 0.05$).

## 3. Results and Discussion

Results from microclimate parameters along with quantitative and qualitative characteristics of the fruits in the two greenhouses are presented below. Specifically, these are air temperature, illuminance, relative humidity, early and total yield, chlorophyll content and transpiration rate, pH, citric acid %, dry matter %, total sugar concentration, and content of fruit bioactive components (ascorbic acid, lycopene, β-carotene, total carotenoids) and antioxidant capacity.

*3.1. Microclimatic Conditions in the Examined Greenhouses*

Microclimatic parameters inside a greenhouse can significantly affect physiological processes (such as photosynthesis) of plant tissues. These parameters are solar radiation, air temperature/humidity, and concentration of carbon dioxide in the air. Recorded data from these parameters were used as measurable variables by instruments quantifying photosynthesis and chlorophyll content. Therefore, they are essential in order to determine photosynthesis, stomatal conductance, transpiration rate, and leaf chlorophyll content. In Figure 3, the air temperature, illuminance, and relative humidity inside the greenhouse are presented for the total duration of the experiment, since they are the parameters that most influence physiological and chemical processes of tomato. Concerning air temperature, there was no difference between the two greenhouses (Figure 3a). However, the illuminance in the DSSC greenhouse was ca. 20% lower compared to the CONV greenhouse (Figure 3b). The shading effect of the DSSC was expected and, moreover, it is beneficial during the summer months. Higher values of light in the CONV greenhouse enhanced yield and physiological characteristics of tomato plants, while lower values in the DSSC greenhouse had a positive effect on fruit qualitative characteristics.

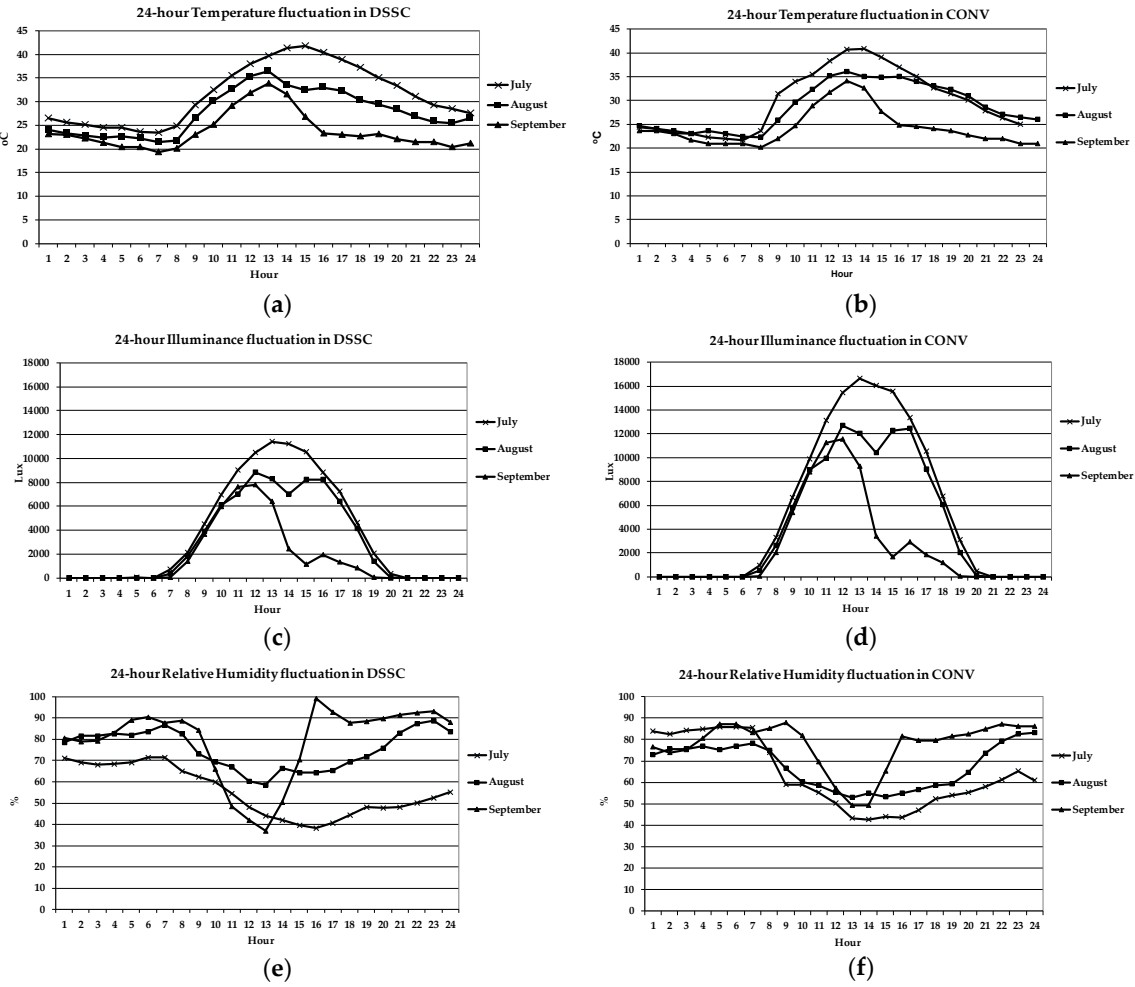

**Figure 3.** Air temperature, illuminance, and relative humidity inside the two greenhouses [CONV (Conventional), DSSC (Dye-Sensitized Solar Cell)]: (**a**) Air temperature (DSSC); (**b**) air temperature (CONV), (**c**) illuminance (DSSC); (**d**) illuminance (CONV); (**e**) relative humidity (DSSC); and (**f**) relative humidity (CONV).

*3.2. Yield and Mean Weight of Early and Total Fruit Production*

The results of early and total production parameters (yield and mean fruit weight) are presented in Figure 4, while a combined analysis of their variance is given in Table 2. Early yield was influenced by the greenhouse type (G), the hybrid species (H), and the G × H interaction, which was also statistically significant (Table 2). In particular, the MST performed differently in the two greenhouses, resulting in a significant yield decrease (by 50.8%) in the DSSC greenhouse (Figure 4a). However, no significant difference was observed in ChT yield in the two greenhouses. The total yield was also influenced by the type of greenhouse and hybrid (Table 2). In particular, for both hybrids a significant yield reduction (by 31% and 40% for ChT and MST, respectively) was observed in the DSSC greenhouse (Figure 4b), possibly due to lower light levels (Figure 3b).

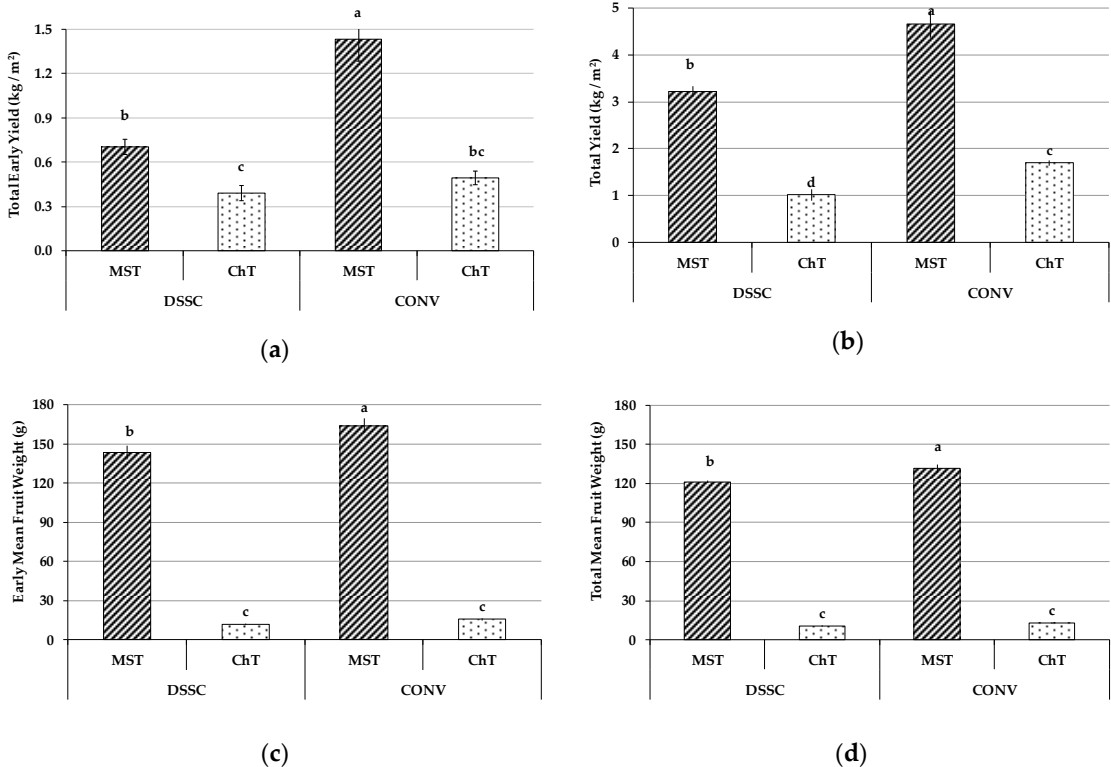

**Figure 4.** Production and yield data of the two tomato hybrids [MST (medium-sized) and ChT (cherry)] in the two greenhouses [DSSC (Dye-Sensitized Solar Cell) and CONV (Conventional)]: (**a**) Early yield; (**b**) total yield; (**c**) early mean fruit weight; (**d**) total mean fruit weight. Bars with different letters were significantly different by LSD at $P \le 0.05$.

**Table 2.** Combined variance analysis of the production and yield data, as influenced by the greenhouse types [DSSC (Dye-Sensitized Solar Cell) and CONV (Conventional)] and the species of tomato hybrid [MST (medium-sized) and ChT (cherry)].

| Source | df [z] | Significance of F Ratio | | | |
|---|---|---|---|---|---|
| | | Early Yield | | Total Yield | |
| | | Early Yield | Fruit Weight | Yield | Fruit Weight |
| Replications | 4 | NS [y] | NS | NS | NS |
| Greenhouse (G) | 1 | ** | NS | ** | ** |
| Error (a) | 4 | | | | |
| Hybrid (H) | 1 | ** | ** | ** | ** |
| G × H | 1 | ** | * | NS | NS |
| Error (b) | 8 | | | | |
| CV % | | 23.87 | 9.30 | 17.59 | 5.64 |

[z] df, degree of freedom; CV, coefficient of variation; [y] NS, nonsignificant; * = $P < 0.05$ level of significance; ** = $P < 0.01$ level of significance.

Average fruit weight in the early production stage was affected by the type of hybrid, but not by the type of greenhouse (Table 2). More specifically, the average fruit weight of MST and ChT was 154 and 14 g, respectively, regardless of the greenhouse (Figure 4c). The MST had a 12.4% higher mean fruit weight in the CONV greenhouse than in the DSSC one, while the ChT fruit weighed the same in both greenhouses. The average fruit weight for total production was also influenced by the type of hybrid and greenhouse (Table 2). The average weight of MST and ChT tomato was 126 and 12 g, respectively, irrespective of the greenhouse (Figure 4d). The MST had a 7.5% higher average fruit

weight in the CONV compared to the DSSC greenhouse, whereas ChT had the same average fruit weight in both greenhouses.

### 3.3. Physiological Parameters of Plants

Results of leaf chlorophyll content, transpiration rate, stomatal conductance, and photosynthetic rate are presented in Figure 5, while a combined analysis of their variance is given in Table 3. The chlorophyll content of leaves (chlorophyll content index, CCI), estimated 40 days after transplanting (DAT), differed between the plants of the two greenhouses and between the hybrids. However, the G × H interaction was not statistically significant. More specifically, in the CONV greenhouse, MST and ChT had a higher CCI than those of the DSSC greenhouse by 37% and 38%, respectively (Figure 5a). The CCI values did not differ between the hybrids grown in the DSSC greenhouse, while they differed significantly in the CONV one. Irrespective of the hybrid, the DSSC greenhouse plants presented lower CCI values compared to plants of the CONV greenhouse.

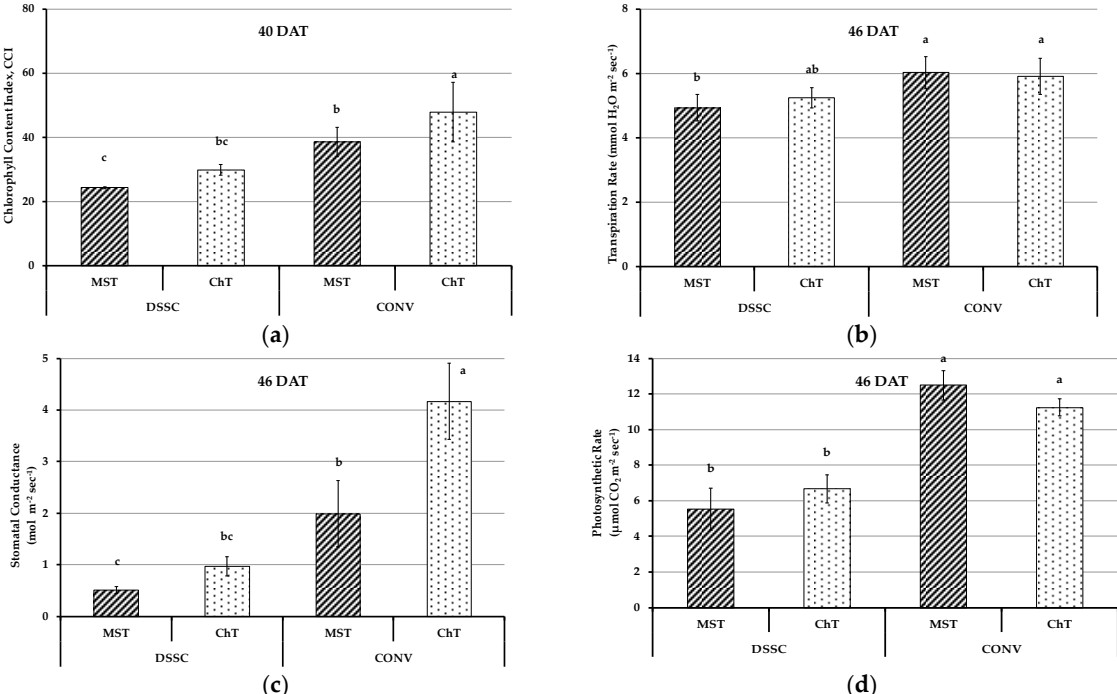

**Figure 5.** Physiological characteristics of the two tomato hybrids [MST (medium-sized) and ChT (cherry)] in the two greenhouses [DSSC (Dye-Sensitized Solar Cell) and CONV (Conventional)] 40 and 46 days after transplanting (DAT): (**a**) Leaf chlorophyll content; (**b**) transpiration rate; (**c**) stomatal conductance; (**d**) photosynthetic rate. Bars with different letters were significantly different by LSD at $P \leq 0.05$.

**Table 3.** Combined variance analysis of the physiological parameters, as influenced by the greenhouse types [DSSC (Dye-Sensitized Solar Cell) and CONV (Conventional)] and the species of tomato hybrid [MST (medium-sized) and ChT (cherry)].

| Source | df [z] | Significance of F ratio | | | |
| --- | --- | --- | --- | --- | --- |
| | | Chlorophyll Content Index | Transpiration Rate | Stomatal Conductance | Photosynthetic Rate |
| | | 40 DAT [y] | 46 DAT | 46 DAT | 46 DAT |
| Replications | 5 | NS [x] | NS | NS | NS |
| Greenhouse (G) | 1 | * | * | ** | ** |
| Error (a) | 5 | | | | |
| Hybrid (H) | 1 | ** | NS | ** | NS |
| G × H | 1 | NS | NS | NS | * |
| Error (b) | 10 | | | | |
| CV % | | 19.38 | 13.31 | 12.71 | 12.74 |

[z] df, degree of freedom; CV, coefficient of variation; [y] DAT, days after transplanting; [x] NS, nonsignificant; * = $P < 0.05$ level of significance; ** = $P < 0.01$ level of significance.

The transpiration rate (measured on day 46) was only influenced by the type of greenhouse (G). The same applied to the G × H interaction (Table 3). The MST in the DSSC greenhouse had an 18% lower transpiration rate compared to the CONV greenhouse (Figure 5b); ChT did not present significant differences between the greenhouses.

Similarly, stomatal conductance differed between the plants of the two greenhouses and between the hybrids, but the G × H interaction was not significant (Table 3). The MST and ChT plants in the CONV greenhouse had higher stomatal conductance (74% and 76%, respectively) compared to the DSSC (Figure 5c).

The photosynthetic rate was mainly influenced by the type of greenhouse (Table 3). The MST and ChT in the DSSC greenhouse had a lower photosynthetic rate (55% and 40%, respectively) compared to the CONV greenhouse (Figure 5d).

Lower values of leaf chlorophyll content, transpiration rate, stomatal conductance, and photosynthetic rate were likely due to the shading effect in the DSSC greenhouse but also to stress from the high air temperature. Yamori et al. [10] and Tewolde et al. [11] reported that supplemental lighting did not lead to higher photosynthesis and yield of tomatoes in summer, when the irradiation from the sun was enough for plant cultivation, whereas it led to higher photosynthesis, growth, and yield in winter when the irradiation is limiting for plant growth. This means that even though light is the basic factor influencing photosynthesis and plant growth, surplus light, especially during summer when it is not needed for plant growth, can be used for electricity generation by photoselective DSSC installed in the greenhouse roof covering [12,13].

### 3.4. Fruit Quality Parameters and Bioactive Compounds

3.4.1. Physicochemical Parameters of Tomato Fruits

In Figure 6a,b, the pH and citric acid percentage of the two tomato hybrids are shown. MST presented significantly higher pH values than ChT, regardless of the greenhouse type (Figure 6a). No significant difference was observed in citric acid content between the two greenhouses for the hybrids. The greenhouse type did not significantly affect dry matter % of either ChT or MST (Figure 6c). Additionally, for total sugar content, no differences were observed between hybrids or greenhouse types (Figure 6d).

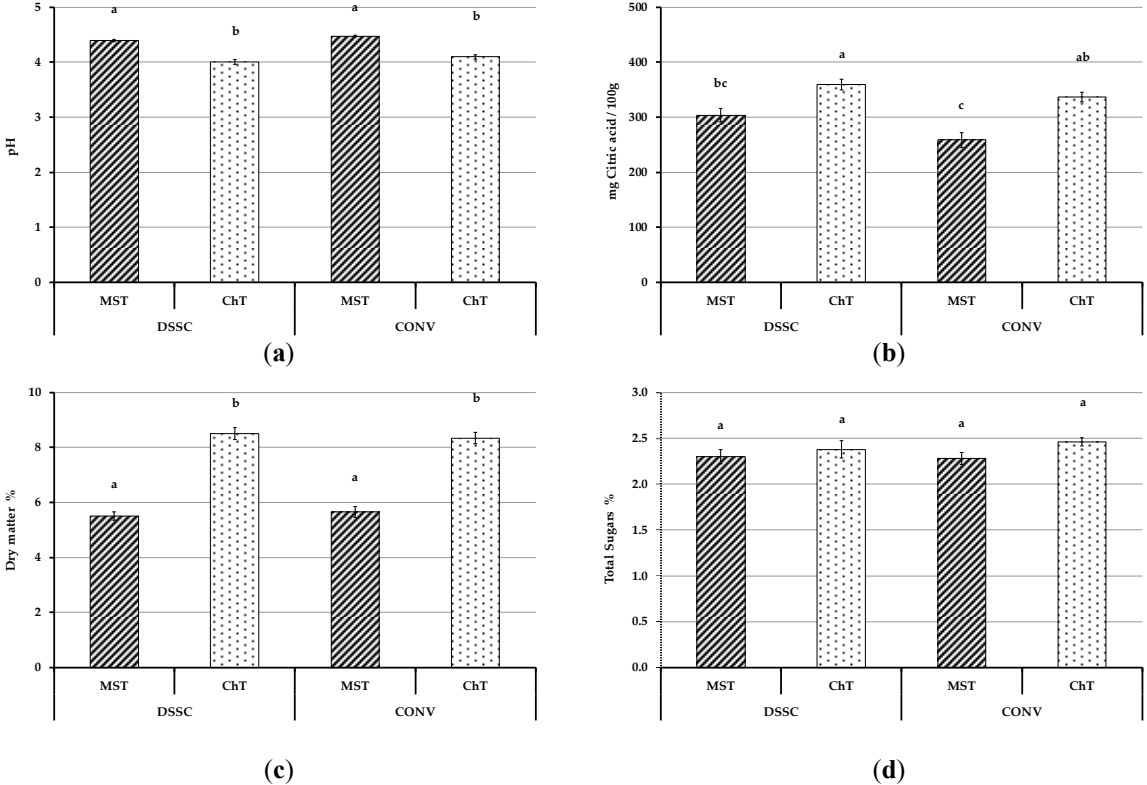

**Figure 6.** Physical parameters of fruits, independent of inflorescence, of the two tomato hybrids [MST (medium-sized) and ChT (cherry)] in the two greenhouses [DSSC (Dye-Sensitized Solar Cell) and CONV (Conventional)]: (**a**) pH; (**b**) citric acid %; (**c**) dry matter %; and (**d**) total sugars %. Bars with different letters were significantly different by LSD at $P \leq 0.05$.

According to the results of the variance analysis (Table 4), the effects of the greenhouse, hybrid, and inflorescence were significant for pH, while the interaction of all three factors was not significant.

**Table 4.** Variance analysis of physical parameters, bioactive ingredients, and antioxidant characteristics, as influenced by the greenhouse type [DSSC (Dye-Sensitized Solar Cell) and CONV (Conventional)] and the species of tomato hybrid [MST (medium-sized) and ChT (cherry)].

| Source | | Significance of F ratio | | | | | | | | |
|---|---|---|---|---|---|---|---|---|---|---|
| | df [z] | pH | CA [y] | SUG | ASC | LUC | β-CAR | CARs | DPPH | ABTS |
| Replications | 2 | NS [x] | NS | NS | NS | NS | NS | NS | ** | ** |
| Greenhouse (G) | 1 | ** | NS | NS | * | ** | * | * | ** | ** |
| Error (a) | 2 | | | | | | | | | |
| Hybrid (H) | 1 | ** | NS | NS | ** | ** | ** | ** | ** | ** |
| G × H | 1 | NS | ** | NS | NS | NS | ** | NS | ** | ** |
| Error (b) | 4 | | | | | | | | | |
| Inflorescence (I) | 4 | ** | ** | ** | ** | ** | ** | ** | ** | ** |
| G x I | 4 | NS | ** | ** | NS | NS | ** | NS | ** | ** |
| H × I | 4 | ** | ** | ** | ** | NS | ** | * | ** | ** |
| G × H × I | 4 | NS | NS | ** | NS | * | * | * | * | ** |
| Error (c) | 32 | | | | | | | | | |
| CV % | | 1.5 | 6.2 | 3.7 | 11.1 | 18.8 | 11.3 | 16.7 | 5.9 | 3.2 |

[z] df, degree of freedom; CV, coefficient of variation; [y] CA, citric acid; SUG; total sugars; ASC, ascorbic acid; LUC, lycopene; β-CAR, β-carotene; CARs, total carotenoids; DPPH, 1,1-diphenyl-2-picrylhydrazyl assay, ABTS, 2,2′-azino-bis-3-ethylbenzthiazoline-6-sulphonic acid assay, [x] NS, nonsignificant; * = $P < 0.05$ level of significance; ** = $P < 0.01$ level of significance.

### 3.4.2. Bioactive Compounds of Tomato Fruits

No significant differences were found between the two greenhouses in ascorbic acid concentration of ChT and MST, but a slight decrease of 6% was observed in the DSSC greenhouse for all tomato samples analyzed (Figure 7a). It is noteworthy that ChT presented a higher ascorbic acid concentration (64%) compared to MST. Several factors contribute to high concentration of ascorbic acid in tomato: variety, high salinity of irrigation water, and solar radiation. Many studies have shown that high light intensity is associated with high vitamin C content in tomatoes [17]. Generally, high temperatures are required for the synthesis of ascorbic acid, and sunlight enhances the accumulation of additional ascorbic acid [24]. It was generally observed that the degree of significance was most affected by the hybrids and less by the greenhouse type, while most of their interactions were not significant (Table 4).

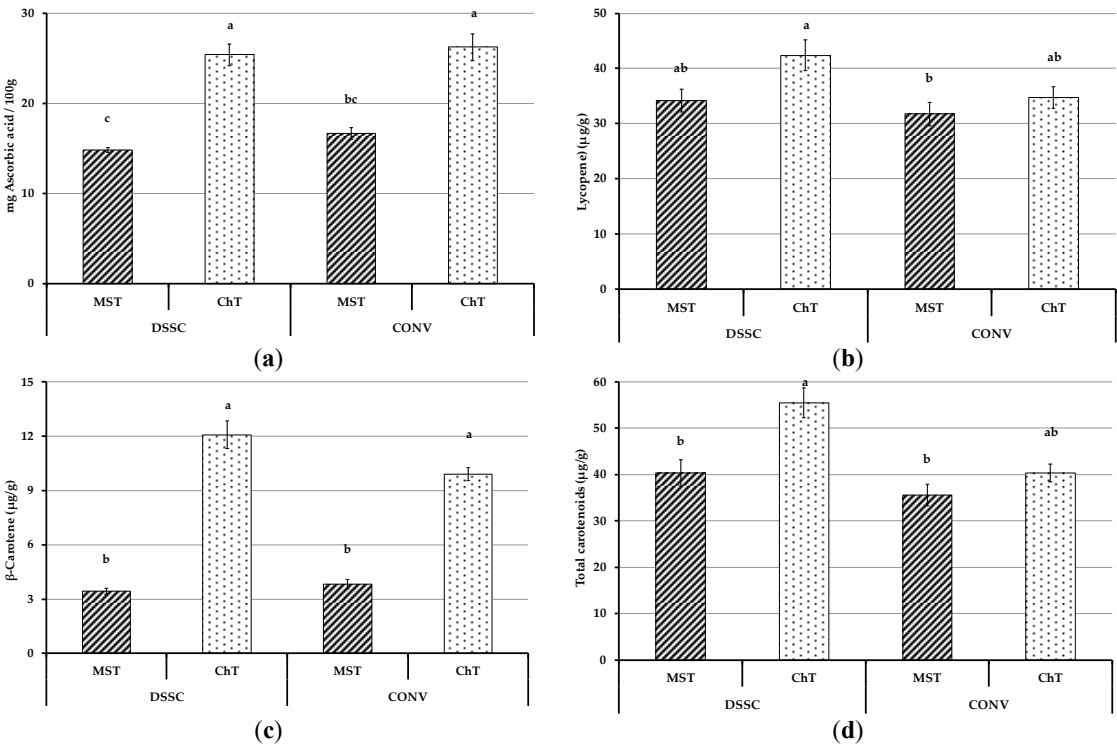

**Figure 7.** Content of bioactive ingredients of the fruits of the two tomato hybrids [MST (medium-sized) and ChT (cherry)] in the two greenhouses [DSSC (Dye-Sensitized Solar Cell) and CONV (Conventional)]: (**a**) ascorbic acid; (**b**) lycopene; (**c**) β-carotene; and (**d**) total carotenoids. Bars with different letters were significantly different by LSD at $P \leq 0.05$.

Concerning the concentrations of lycopene, β-carotene, and total carotenoids, significant differences were observed both between tomato hybrids and between inflorescences and greenhouses, as well as their interactions (Table 4). ChT presented higher values of β-carotene (11.0 μg/g) compared to MST (3.6 μg/g) in both greenhouses, whereas ChT had the higher value of lycopene (42.4 μg/g) in the DSSC greenhouse. In the DSSC greenhouse there was a 15% and 13% increase in the concentration of lycopene and β-carotene, respectively, compared to the CONV greenhouse (Figure 7b,c). Similar results were reported by Pek et al. [24], who found that tomatoes that were shaded had a higher concentration of lycopene. Similarly to β-carotene, ChT showed higher total carotenoid concentration (48 μg/g) compared to MST (38 μg/g) in both greenhouses (Figure 7d). In general, an increase of about 26% in concentration of total carotenoids was observed in the DSSC greenhouse for both hybrids.

### 3.4.3. Antioxidant Capacity

Concerning the antioxidant capacity, significant differences were observed between greenhouses, tomato hybrids, inflorescences, as well as their interactions (Table 4). For both hybrids, the ABTS and DPPH tests showed increases of about 10% and 5% in the DSSC greenhouse (Figure 8a,b). It is worth noting that the antioxidant capacity of the DPPH test was lower than ABTS, because the DPPH test was sensitive to obstructions due to the high concentration of carotenoids.

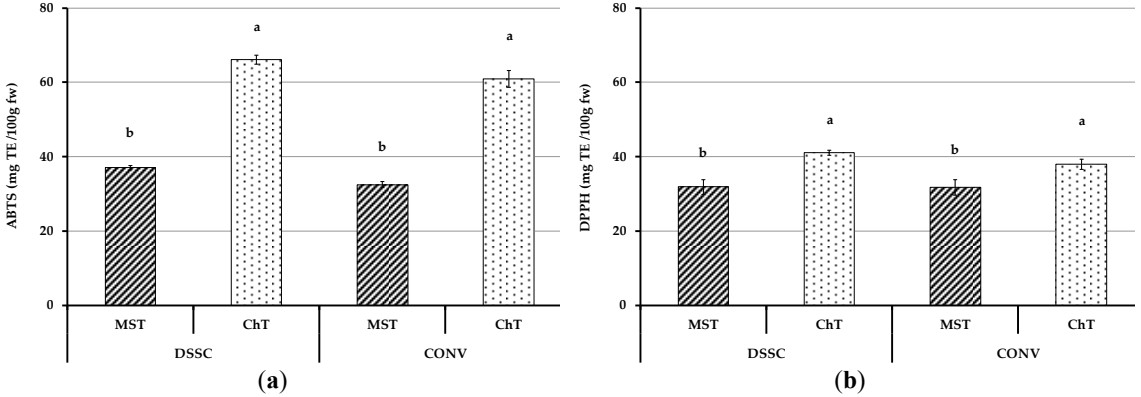

**Figure 8.** Antioxidant capacity of the fruits of the two tomato hybrids [MST (medium-sized) and ChT (cherry)] in the two greenhouses [DSSC (Dye-Sensitized Solar Cell) and CONV (Conventional)]: (**a**) ABTS (2,2′-azino-bis-3-ethylbenzthiazoline-6-sulphonic acid) assay; and (**b**) DPPH (1,1-diphenyl-2-picrylhydrazyl) assay. Bars with different letters were significantly different by LSD at $P \leq 0.05$.

## 4. Conclusions

Data concerning crop productivity and physiology were obtained through the hydroponic cultivation of tomato crops in two experimental greenhouses, one using the DSSC technology as covering material and a conventional one covered by glass as a control. Particular attention was given to the physiological response of the crops under the novel covering material, which resulted in a shading effect. For this reason, specific physiological parameters of the crop, such as plant transpiration rate and leaf chlorophyll content, were measured and monitored along with potential photomorphogenetic responses of the plants. Furthermore, the comparisons included chemical analyses of harvested tissues and fruits in order to determine potential differences in the concentration of phytochemical compounds, such as ascorbic acid, lycopene, and quality characteristics. The early yield and average fruit weight of the MST were affected by the DSSC greenhouse, but not the ChT. The total yield and average fruit weight of the MST were also affected by the DSSC greenhouse. As for the ChT, its total yield was affected by the DSSC greenhouse, but not average fruit weight. The DSSC greenhouse plants presented lower CCI and transpiration rate values, especially for the MST, compared to plants in the CONV greenhouse. However, the transpiration rate of the ChT did not present statistically significant differences between the two greenhouses. Additionally, the DSSC greenhouse showed significantly higher values of dry matter % than the CONV for both the ChT and MST, while no differences in the total sugar content were observed for the greenhouses. An increase of about 20% in concentrations of lycopene and β-carotene was observed in the DSSC greenhouse for the ChT and 10% for the MST. Finally, results showed that the ABTS test is the more appropriate method for measuring antioxidant capacity in tomato extracts. The results from the DSSC greenhouse during the summer season were satisfactory, since shading had a positive effect on the qualitative characteristics of the tomato fruits; however, lower values of physiological characteristics and yield were related to stress from high air temperatures. Furthermore, surplus light, especially during summer when it is not needed for plant growth, can be used for electricity generation by photoselective DSSCs installed in the greenhouse roof

covering. In future work, the DSSC greenhouse will be evaluated during the entire year, especially during wintertime.

**Author Contributions:** Methodology, G.K.N., K.K. (Kalliopi Kadoglidou), N.T. and K.K. (Konstantinos Krommydas); investigation, K.K. (Kalliopi Kadoglidou), N.T. and K.Kr. (Konstantinos Krommydas); data curation, K.K. (Kalliopi Kadoglidou), K.K. (Konstantinos Krommydas) and M.I.; formal analysis, K.K. (Kalliopi Kadoglidou); writing—original draft preparation, G.K.N. and K.K. (Kalliopi Kadoglidou); writing—review and editing, G.K.N., K.K. (Kalliopi Kadoglidou), A.K., P.R. and M.I.

**Funding:** This research was funded by the Operational Programme "Competitiveness and Entrepreneurship", National Action "Cooperation 2011–Research Project 11 SYN _7 _298 "Energy autonomous smart greenhouse".

**Conflicts of Interest:** The authors declare no conflicts of interest.

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
