# Peer review of "Performance and Hydroponic Tomato Crop Quality Characteristics in a Novel Greenhouse Using Dye-Sensitized Solar Cell Technology for Covering Material"

_horticulturae, doi:10.3390/horticulturae5020042_

Round 1

Reviewer 1 Report

While the results of Ntinas et al. are fully within the scope of the journal, I have some comments as followings:

Please revise title. It is not clear:

suggestion: Hydroponic tomato performances using Dey-Sensitized Solar Cell for shading greenhouse

The authors should not report the not significant results both in the abstract and in the manuscript.

Why the authors used tomato as crop in this study?

A critical point is that the experiment is performed just in one crop cycle.

Please introduce all acronyms/abbreviations at first appearance and then use them consistently through the manuscript.

Lines 55-57. The sentence in son not clear. Please rewrite the sentence.

Line 59. ….tomato). °Brix……

Line 63. …..bioactive compounds (which ones?)

Line 70. ….bioactive compounds with antioxidant activity…(which ones?)

Line 73-74. The authors can remove the sentence. The same concept was already reported above.

Line 96. ……. is changing during maturation (the authors should report the main changes)

Figure 1 is not essential in this manuscript. Rather, I suggest to include a figure showing the differences in the greenhouse shading.

Lines 123-125. Please rewrite the sentence, it is not clear. Moreover, the authors should report the corresponding stage of growth.

Which instruments were used to measure the main climatic parameters?

Why the authors did not record important traits such as total biomass (leaf + stem + root + fruit), LAI, etc.? This data is essential for this study.

Why the authors report only information on transpiration? Also stomatal conductance, net assimilation are two important and fundamental parameters in this study.

For the used instrument the authors should report the name of the company and its location.

Line 145. 1/3? What is the meaning?

Line 150. Slurries is the correct word in this context?

Line 154. Why 72°C? Why 48 h? Did you reach the constant weight?

Line 160. ……of tomato pulp (the authors should add a reference)

The authors should report the agronomic practices such as, the commercial name of the rockwool, the amount and the type of fertilizer used for the cultivation, the volume of water used during the crop growth, the name of the insecticides and fungicides applied, etc.. This information is fundamental in this manuscript.

The authors do not provide any information, whether the assumptions of the statistical test (ANOVA) were met. These include homogeneity of variance and normality of the data for all Parameters and groups. were these assumptions tested and confirmed?

I am missing the discussion part where the explanations are given why these results are observed. The authors should describe the motivation of the obtained results. Moreover, the authors should compare their results with data available in the literature. So what is special about DSSC for shading the greenhouse?

Be careful that when two treatments are a common letter such as T1 a and T1 ab the two treatments are similar and not different. This kind of mistakes is frequent in the sections 3.4.2. and 3.4.3, the authors should rewrite these sections.

Line 321-324. This sentence should be moved into the section 2.3.

Conclusions. You need to discuss the influence of DSSC and how this affects the hydroponic tomato performances and thus again its relevance for shading greenhouse.

I hope that my comments and suggest can help you to improve your manuscript.

Author Response

Dear Editor and Reviewers,

We appreciate the constructive and detailed remarks of the three reviewers of the manuscript horticulturae-478842 “Performance and hydroponic tomato crop quality characteristics in a novel greenhouse using DSSC technology for shading”. We gladly faced the reviewers’ comments since this will undoubtedly improve our work. We appreciate their laborious evaluation. In addition to the revisions based on the reviewers’ comments, minor modifications where necessary.

Please find at the attached file our reply to your review report. 

Best Regards

Dr. Kadoglidou Kalliopi

Reviewer 2 Report

see attached file for ediorial corrections and comments.

Author Response

(The authors gave the same response as above.)

Reviewer 3 Report

The present study compared plant growth, quantity and quality of tomatoes between Dye-Sensitized Solar Cell (DSSC) technology and a conventional one (CONV) covered by diffusion glass as a control. A lot of hard work has been put into the study and thus the work includes some attractive points. However, it does not meet the basic international standards in data presentation. For example, it is highly descriptive without appropriate references and the style of the writing is similar more and more to a report. Moreover, I feel that the fundamental analyses are incomplete. Below I have outlined points that I think could improve the MS, since all of sections needs to be re-written to improve whole manuscript.

1) Paper format and English check

(1) This manuscript is FULL PAPER, but Results and Discussion is combined. I do not know the general format in this Journal, but they should be separated if needed. In addition, as described above, the current style of the writing is similar to a simple report and does not include meaningful discussion based on the data set. Therefore, I recommend the authors to discuss something like a recent study which used various DSSC technology for covering materials and a light-blocking curtain. In addition, the authors should analyze the electricity generation and discuss its benefit. Even if the yield and quality could be worsened in DSSC greenhouse, the net income of the DSSC technology could be higher than that of CONV. I think that this is the most important part in this manuscript. If it is hard to analyze them, please describe their possibility in Discussion.

(2) English should be checked by a native English speaker.

2) Table & Figure preparations with appropriate statistical analyses

I recommend the authors to examine statistical analysis between DSSC and CONV in MST and ChT, respectively, since there are no meaning to examine statistical analysis between MST and ChT. Accordingly, I recommend the authors to show Figures in MST and ChT with different Y axis, separately.

3) Photosynthetic measurement

It seems that gas-exchange was measured in a portable system, but only transpiration rate was shown in a Figure. Why do not you show photosynthetic rate and stomatal conductance? In addition, please describe what conditions the authors measured gas-exchange (e.g., light intensity, CO2 conc., temperature, humidity).

4) Environmental data set

Please describe DLI anywhere. I’m wondering why the light intensity is so low. Have you measured light intensity outside? Is the value general?

Since light intensity is lower in DSSC than in CONV, the temperature inside the greenhouse in DSSC should be lower. However, the greenhouse temperature in July and August is similar between DSSC and CONV, and in September is higher in DSSC than in CONV. It is very unusual. I could not see any description for environmental control for plant cultivation in the present study. Please describe an environmental control in M&M, and mention the reason why the greenhouse temperature is higher in DSSC than in CONV.

By the way, I worry that the temperature in a glasshouse is very high (i.e., about 40C) in the present study. As you know, high temperature decrease photosynthesis and thus plant growth (e.g., Yamori et al. 2010 Plant Physiology 152, 388-399; Tewolde et al. 2016, Frontiers in plant science, 7, 448). Therefore, it would be better to mention that high temperature stress could affect the tomato quality in DSSC in the present study.

(5) The cited literature needs to be revised to include more appropriate and recent studies.

For example,

L199, please cite recent papers by Yamori 2016 (Journal of Plant Research 129:379–395) and Yamori et al. 2016 (Plant Cell Environ. 39:80–87), since these papers summarized that various environmental factors could affect photosynthetic reaction and showed a clear relationship between photosynthesis and crop yield.

In discussion (and also Introduction), the advances and benefits in DSSC technology could be described. For example, the authors could cite the following papers: Roslan et al. (2018) Renewable and Sustainable Energy Reviews, 92, 171-186; Prabavathy et al. (2017) International Journal of Energy Research, 41(10), 1372-1396; Khan et al. (2016) Renewable and Sustainable Energy Reviews, 55, 414-425.

In addition, it would be good to cite the following recent paper and discuss: for example, Tewolde et al. (2016, Frontiers in plant science, 7, 448) who showed that supplemental lighting did not lead to higher photosynthesis and yield of tomatoes in summer when the irradiation from the sun is enough for plant cultivation, whereas it led to higher photosynthesis, growth, and yield in winter when the irradiation is limiting for plant growth. It means that light is basically important for photosynthesis and plant growth (Yamori et al. 2016 Plant Cell Environ. 39:80–87), but the excessive light is not needed for plant growth, leading that the excessive light can be used for DSSC for electricity generation. Then, the authors can expand the discussion effect of DSSC on plant growth during the whole year.

Author Response

(The authors gave the same response as above.)

Round 2

Reviewer 1 Report

The manuscript has been improved following Reviewer suggestions. However, please add the chemical characteristics of the two nutrient solution used in the experiment (in term of N, P, K, EC, pH e micronutrients)

Author Response

Response to Reviewer 1 Comments

Open Review

English language and style

( ) Extensive editing of English language and style required 
( ) Moderate English changes required 
( ) English language and style are fine/minor spell check required 
(x) I don't feel qualified to judge about the English language and style 

Yes

Can be improved

Must be improved

Not applicable

Does the introduction   provide sufficient background and include all relevant references?

(x)

( )

( )

( )

Is the research design   appropriate?

( )

(x)

( )

( )

Are the methods adequately   described?

(x)

( )

( )

( )

Are the results clearly   presented?

( )

(x)

( )

( )

Are the conclusions   supported by the results?

(x)

( )

( )

( )

Comments and Suggestions for Authors

We thank the reviewer for his/her appreciation of our work. Please find below the response to your comment.

Point 1:

The manuscript has been improved following Reviewer suggestions. However, please add the chemical characteristics of the two nutrient solution used in the experiment (in term of N, P, K, EC, pH e micronutrients)

Response 1:

Done as the reviewer suggested. In the new version we added the relative information in M&M section, 2nd paragraph of subsection 2.1:

Tomato crops in both greenhouses were fertirrigated with Hoagland's solution (Table 1).

Table 1. Chemical characteristics of the nutrient solutions used in the experiments.

Stock solution

(Tank A)

Stock solution

(Tank B)

Micronutrients

(Tank C)

Macronutrients

(g/100L)

Macronutrients

(g/100L)

Micronutrients

(g/20L)

Ca(NO3)2

6800

K2SO4

2780

MnSO4

585

KNO3

1000

KNO3

1320

Na2B4O7

530

EDTA-Fe (13%)

200

MgSO4

1300

CuSO4

40

KH2PO4

2340

ZnSO4

270

(NH4)6Mo7O24

25

Quantity of 2.8 L of HNO3 where added in a fourth tank containing 100L of water. The pH and the EC of the nutrient solution were 5.5 - 6.0 and 3.0 - 3.5 dS m-1, respectively. The EC and pH were measured using portable EC and pH meters (HI 8733 and HI 8424, Hanna Instruments, Inc., Woonsocket, RI, USA).

Submission Date

21 March 2019

Date of this review

16 Apr 2019 11:15:27

Reviewer 2 Report

this version reads a lot better. thanks for spending the time and effort to update it.

Author Response

Response to Reviewer 2 Comments

Open Review

English language and style

( ) Extensive editing of English language and style required 
( ) Moderate English changes required 
(x) English language and style are fine/minor spell check required 
( ) I don't feel qualified to judge about the English language and style 

Yes

Can be improved

Must be improved

Not applicable

Does the introduction   provide sufficient background and include all relevant references?

(x)

( )

( )

( )

Is the research design   appropriate?

( )

( )

( )

( )

Are the methods adequately   described?

(x)

( )

( )

( )

Are the results clearly   presented?

(x)

( )

( )

( )

Are the conclusions   supported by the results?

(x)

( )

( )

( )

Comments and Suggestions for Authors

Point 1: this version reads a lot better. thanks for spending the time and effort to update it.

Response 1: We thank the reviewer for his/her appreciation of our work.

Submission Date

21 March 2019

Date of this review

16 Apr 2019 16:28:29

Reviewer 3 Report

The manuscript has been revised well. I still have some comments. Please edit more according to the following comments.

Comment 1

I still feel that scientific English is not used so often, although it seems that English was checked by a native English speaker, according to the authors. Please describe the person’s name or company in Acknowledgement.

Comment 2

P1 L 34: Add the following sentence to the make sentences logical. Since plants in natural environments must cope with diverse, highly dynamic, and unpredictable conditions (Yamori 2016, Journal of Plant Research 129:379–395), such systems provide a shelter for the crop against xxxxxxxxxxxx.

Comment 3

P2 L 92-106: I think that it is good to introduce the role of lycopene and other carotenoids in Introduction. However, too many information. Please reduce this paragraph by half. In addition, please add appropriate references each sentence (P 2 L 93-97).

Comment 4

P4 L278~: There are several types of Hoagland's solution. Please specify it.

P4 L302~: It is so surprising that the relative humidity is extremely low… Please show diurnal change of relative humidity in Figure 3.

Comment 5

P 4 L 281~: Environmental data, including solar radiation, has been recorded with a HOBO data logger. Thus, DLI can be calculated. Please describe the data somewhere.

Comment 6

P9 Figure 5c: The unit of Figure 5c is wrong. Please check it again.

Comment 7

Abstract and P13 L 693~: The authors cannot conclude that ABTS test was more appropriate method for measuring antioxidant capacity in tomato extracts, based on the data set in this manuscript. I recommend the authors to delete it.

Author Response

Response to Reviewer 3 Comments

Open Review

English language and style

(x) Extensive editing of English language and style required 
( ) Moderate English changes required 
( ) English language and style are fine/minor spell check required 
( ) I don't feel qualified to judge about the English language and style 

Yes

Can be improved

Must be improved

Not applicable

Does the introduction   provide sufficient background and include all relevant references?

( )

( )

(x)

( )

Is the research design   appropriate?

(x)

( )

( )

( )

Are the methods adequately   described?

( )

(x)

( )

( )

Are the results clearly   presented?

( )

(x)

( )

( )

Are the conclusions   supported by the results?

( )

(x)

( )

( )

Comments and Suggestions for Authors

The manuscript has been revised well. I still have some comments. Please edit more according to the following comments.

We thank the reviewer for his/her appreciation of our work. Please find below a point by point response to each of your comments.

Point 1: Comment 1

I still feel that scientific English is not used so often, although it seems that English was checked by a native English speaker, according to the authors. Please describe the person’s name or company in Acknowledgement.

Response 1: The authors revised the scientific English furthermore. Unintentionally, we wrote, taking the phrase of the reviewers comment that a native English speaker checked the manuscript.

Point 2: Comment 2

P1 L 34: Add the following sentence to the make sentences logical. Since plants in natural environments must cope with diverse, highly dynamic, and unpredictable conditions (Yamori 2016, Journal of Plant Research 129:379–395), such systems provide a shelter for the crop against xxxxxxxxxxxx.

Response 2: The authors thank the reviewer for the suggestion, however the suggested phrase is rather confusing than logic to us, and it is not related to the topic of this paper.

Point 3: Comment 3

P2 L 92-106: I think that it is good to introduce the role of lycopene and other carotenoids in Introduction. However, too many information. Please reduce this paragraph by half. In addition, please add appropriate references each sentence (P 2 L 93-97).

Response 3: Done as the reviewer suggested.

Point 4 (a and b): Comment 4

Point 4a: P4 L278~: There are several types of Hoagland's solution. Please specify it.

Response 4a: Done as the reviewer suggested. In the new version we added the relative information in M&M section, 2nd paragraph of subsection 2.1:

Tomato crops in both greenhouses were fertirrigated with Hoagland's solution (Table 1).

Table 1. Chemical characteristics of the nutrient solutions used in the experiments.

Stock solution

(Tank A)

Stock solution

(Tank B)

Micronutrients

(Tank C)

Macronutrients

(g/100L)

Macronutrients

(g/100L)

Micronutrients

(g/20L)

Ca(NO3)2  

6800

K2SO4

2780

MnSO4

585

KNO3

1000

KNO3

1320

Na2B4O7

530

EDTA-Fe (13%)

200

MgSO4

1300

CuSO4

40

KH2PO4

2340

ZnSO4

270

(NH4)6Mo7O24

25

Quantity of 2.8 L of HNO3 where added in a fourth tank containing 100L of water. The pH and the EC of the nutrient solution were 5.5 - 6.0 and 3.0 - 3.5 dS m-1, respectively. The EC and pH were measured using portable EC and pH meters (HI 8733 and HI 8424, Hanna Instruments, Inc., Woonsocket, RI, USA).

Point 4b: P4 L302~: It is so surprising that the relative humidity is extremely low… Please show diurnal change of relative humidity in Figure 3.

Response 4b: In fact this value (35 ± 3 mBar) is the water reference as partial pressure (ranges 0-75) and it was recorded inside the leaf chamber of the portable photosynthesis system. We changed it in M&M, subsection 2.2.2, with the appropriate term. However, the RH values of Hobo micro-station indicated that RH ranged by 37 to 99% in DSSC, and by 43 to 88% in CONV greenhouse. As the reviewer suggested, we added the 24-hour relative humidity fluctuation in DSSC and CONV greenhouse during experimentation in Figure 3.

Point 5: Comment 5

P 4 L 281~: Environmental data, including solar radiation, has been recorded with a HOBO data logger. Thus, DLI can be calculated. Please describe the data somewhere.

Response 5: Solar radiation, has been recorded with a HOBO data logger, however the data set does not contain all data. Unfortunately, it is not possible for the authors to provide DLI values.

Point 6: Comment 6

P9 Figure 5c: The unit of Figure 5c is wrong. Please check it again.

Response 6: That was a mistake. The unit is revised as the reviewer suggested. Thank you very much for noticing!

 Point 7: Comment 7

Abstract and P13 L 693~: The authors cannot conclude that ABTS test was more appropriate method for measuring antioxidant capacity in tomato extracts, based on the data set in this manuscript. I recommend the authors to delete it.

Response 7: Done as the reviewer suggested.

Submission Date

21 March 2019

Date of this review

16 Apr 2019 08:47:55

Round 3

Reviewer 3 Report

The manuscript has been revised well. Please edit more according to the following comments.

P1 L 34: I still think that the following citation would be much better.

Such systems provide a shelter for the crop against the direct influence of external weather conditions (Yamori 2016, Journal of Plant Research 129:379–395) xxxxxxxxx.

Figure 5c: I still think that the unit is wrong.....

Author Response

We thank the Reviewer for his/her notes. Please find below a point by point response to each of your comments.

Comments and Suggestions for Authors

The manuscript has been revised well. Please edit more according to the following comments.

P1 L 34: I still think that the following citation would be much better.

Such systems provide a shelter for the crop against the direct influence of external weather conditions (Yamori 2016, Journal of Plant Research 129:379–395) xxxxxxxxx.

Done as the Reviewer suggested.

Figure 5c: I still think that the unit is wrong.....

We double-checked the units in Figure 5c and these are correct, according to the LCi-SD user guide.